# Evaluation of Chemical Composition among the Multi Colored Germplasm of *Abrus precatorius* L.

**DOI:** 10.3390/plants13141963

**Published:** 2024-07-18

**Authors:** Prabakaran Sampath, Sowmyapriya Rajalingam, Sharmila Murugesan, Rakesh Bhardwaj, Veena Gupta

**Affiliations:** 1The Graduate School, ICAR-Indian Agricultural Research Institute, Pusa Campus, New Delhi 110012, India; sowmiyapriya97@gmail.com (S.R.); sharmi95pgr@gmail.com (S.M.); 2Division of Germplasm Evaluation, National Bureau of Plant Genetic Resources, New Delhi 110012, India; 3Division of Germplasm Conservation, National Bureau of Plant Genetic Resources, New Delhi 110012, India; veena.gupta@icar.gov.in

**Keywords:** *Abrus precatorius*, anthocyanin, flavonol, antioxidative potential, medicinal herb

## Abstract

The medicinal plant *Abrus precatorius* L. was traditionally used in the Siddha and Ayurvedic systems of medicine in India. The Indian center of origin holds a vast variability in its seed color. The objective of this study was to assess the total monomeric anthocyanin, flavonol, as well as the antioxidative potential, protein content and ash content among the accessions. A total of 99 accessions conserved in the Indian National Genebank were used in this study. The methods used for the estimation of total monomeric anthocyanin, flavonol, as well as the antioxidative potential, protein content and ash content were the pH differential method, Oomah method, Ferric Reducing Antioxidant Potential, Dumas method and gravimetric method, respectively. The completely black colored accession was recorded with highest total monomeric anthocyanin (51.95 mg/100 g of cyanidin 3-glucoside equivalent) and flavonol content (66.41 mg/g of quercetin equivalent). Red + black colored accessions have recorded the maximum value with respect to antioxidants (14.18 mg/g of gallic acid equivalent). The highest amount of protein content was found in a completely white colored accession (20.67%) and the maximum ash content was recorded in red + black colored accession (4.01%). The promising accessions identified can be used by pharmaceutical companies in drug development and in curing degenerative diseases.

## 1. Introduction

*Abrus precatorius* L. is a medicinal herb native to the Indian subcontinent. The genus name *Abrus* was derived from Latin, which denotes “graceful” or “beautiful” [1]. It connotes the attractive appearance of its seed. The vast genetic variability of the species occurs throughout the country. It was commonly known as Indian liquorice, and belongs to the family fabaceae. However, the plant was known by different names in various parts of the world, such as Gunja, Rosary pea, Crab’s eye, and Jequerity bean [2]. In the outer Himalyan region, it was found at an altitude of up to 1200 m above sea level [3]. Due to the presence of toxic albumin abrin, the seeds should not be consumed without detoxification [4]. Even a very small quantity of abrin (0.1–1 μg/kg) can cause death in humans [5]. In the Ayurvedic texts of India, various detoxification techniques were available. The process was called “Shodhana”, which denotes the removal of the toxic chemical compound without affecting its efficacy in curing diseases [6].

In the traditional system of medicine, the plant was known for the treatment of various diseases or health problems [3], such as rheumatic arthritis and dysentery, and it was also used as an aphrodisiac and an abortifacient [2,3,7,8]. It was observed that the seed kernel was mainly comprised of protein, but the oil and starch content were low. The seeds were rich in mineral nutrients like phosphorous, sulfur, potassium, calcium, magnesium and iron. The entire plant (roots, leaves and seeds) was used in the Siddha, Unani and Ayurvedic systems of medicine [7]. The previous workers have reported that the seed extract was found to possess anti-cancer activity [9]. In ethanobotany, the leaf extract from Abrusosides A–D, four novel sweet-tasting triterpene glycosides from the leaves of *Abrus precatorius*, was known for curing rabies, tetanus, cat and dog bites [3]. The sweet taste of the leaves was due to the presence of these compounds, which means it is 30 to 100 times sweeter than that of sucrose [10]. In folk medicine, a paste of the leaves or root was used as a remedy for a snake bite. The root powder can also be applied topically [11]. The leaves were found to be the ideal material for the production of silver nanoparticles, and possess good antibacterial activity [12]. The detoxified seeds (by boiling) of *A. precatorius* are eaten as food by the residents of Andaman and Nicobar Islands in India. The Onges tribe consumes them when food is scarce. Ratti seeds are rich in essential amino acids, except cysteine and threonine [13]. The calorific value of *Abrus* seeds is better in terms of food energy value than any other fabaceae seeds. However, the heat labile antinutritional components have to be removed by proper cooking [7]. The Indian center of origin represents the vast genetic variability for seed color. The previous studies, which were mentioned above, were carried out with only very few accessions. However, the present experiment was carried out with 99 accessions, collected from different regions in India. Also, none of the earlier researchers have conducted the experiments by including the entire variability for the seed color present in the region. Most of them focused only on red + black seed color combinations. However, various other seed color accessions were not included in their study. Therefore, the aim of the present study was to investigate the differences in the chemical constituents among the multi seed colored *Abrus* germplasm.

## 2. Results

### 2.1. Total Monomeric Anthocyanin (mg/g of QE)

From Table 1, it was observed that highly significant variation exists at a 1% level of significance. Furthermore, a pairwise comparison of the means of all the accessions through least significant difference (LSD) indicated significant differences among them, with the critical difference value of 8.06. The observed mean value of total monomeric anthocyanin among the Ratti accessions studied was 23.04 ± 1.22 mg/100 g of CGE, and the median value was 27.07 mg/100 g of CGE (Table 2). The samples were recorded with a range of 51.95 mg/100 g of CGE. The completely black colored accession was reported with the maximum value of total monomeric anthocyanin, i.e., 51.95 mg/100 g of CGE, whereas it was absent in complete white, cream + brown and pink + brown accessions. The top three accessions that recorded the highest amount of total monomeric anthocyanin were IC0605143 (51.95 mg/100 g of CGE), IC0405311 (49.83 mg/100 g of CGE) and IC0401666 (47.63 mg/100 g of CGE), as given in Table 3. All three of these accessions had the black seed coat color. The mean value of the total monomeric anthocyanin (mg/100 g of CGE) among different seed colored *Abrus precatorius* accessions is shown in Figure 1.

### 2.2. Flavonols (mg/g of QE)

From Table 1, it can be observed that highly significant variation exists at a 1% level of significance. Furthermore, a pairwise comparison of the means of all the accessions through least significant difference (LSD) indicated significant differences among them, with the critical difference value of 12.75. The average amount of flavonol content among the 99 Ratti accessions studied was 42.23 ± 0.83 mg/g of QE, and the median value was 44.04 mg/g of QE. The accessions were recorded with a range of 40.80 mg/g of QE (Table 2). The completely white colored accession was recorded with the lowest amount of flavonols, i.e., 25.61 mg/g by IC0545109, whereas the completely black colored accession was recorded with the highest amount of flavonols, i.e., 66.41 mg/g by IC0405311. From Table 3, it can be observed that the top three accessions which were observed with the highest amount of flavonols were IC0405311 (66.41 mg/g), IC0401666 (64.31 mg/g) and IC0605143 (59.91 mg/g). All three of these accessions which were recorded with the highest amount of flavonols had a black seed coat color. The mean value of flavonol (mg/g of QE) among different seed colored *Abrus precatorius* accessions is shown in Figure 2.

### 2.3. Antioxidants (mg/g of GAE)

One of the important applications of *Abrus* in the pharmaceutical industry made use of its antioxidant property [14,15]. From Table 1, it can be observed that highly significant variation exists at a 1% level of significance. Furthermore, a pairwise comparison of the means of all the accessions through least significant difference (LSD) indicated significant differences among them, with the critical difference value of 0.76. Among the accessions studied, the mean value of antioxidant was 10.24 ± 0.45 mg/g of GAE, and the median value of antioxidant was 11.65 mg/g of GAE (Table 2). Its range in the accessions studied was 13.42 mg/g of GAE. The minimum value with respect to antioxidants was recorded for the white colored Ratti accessions, i.e., 0.76 mg/g of GAE by IC0545109. On the other hand, the maximum value with respect to antioxidants was recorded for the red + black color combination of Ratti accessions, i.e., 14.18 mg/g of GAE by IC0405295. From Table 3, it can be observed that the top three accessions with the highest amount of antioxidants were IC0405295 (14.18 mg/g), IC0385638 (14.13 mg/g) and IC0405311 (14.07 mg/g). The mean value of antioxidants (mg/g of GAE) among different seed colored *Abrus precatorius* accessions is shown in Figure 3.

### 2.4. Protein Content

The boiled seeds were consumed as food by the Onges tribe in India [13]. From Table 1, it can be observed that highly significant variation exists at a 1% level of significance. Furthermore, a pairwise comparison of the means of the all accessions through least significant difference (LSD) indicated significant differences among them, with the critical difference value of 3.84. Among the accessions studied, the average value of protein content was 17.99 ± 0.11%, and the median value of protein content was 18.03% (Table 2). Its range among the accessions studied was 5.23%. The minimum value of protein content was recorded by the completely white colored Ratti accession, i.e., 15.43% by IC0392840. The highest amount of protein content (%) was also found in the complete white colored accession, i.e., IC0385644 (20.67%). From Table 3, it can be observed that the top three accessions with the highest amount of protein content were IC0385644 (20.67%), IC0538733 (20.41%) and IC0349819 (20.17%). The mean value of protein content (%) among different seed colored *Abrus precatorius* accessions is shown in Figure 4.

### 2.5. Ash Content

From Table 1, it can be observed that highly significant variation exists at a 1% level of significance. Furthermore, a pairwise comparison of the means of all the accessions through least significant difference (LSD) indicated significant differences among them, with the critical difference value of 1.71. The ash content of *Abrus* sample helps in estimating the amount of inorganic residues left after ignition. Based on the analysis of ash content in *Abrus* seeds, it was found that the mean value was 3.28 ± 0.04%, and the median value was 3.34% (Table 2). The range was estimated as 1.93%. The minimum value of ash content was recorded for the red + black seed colored Ratti accession, i.e., 2.08% by IC0310646. The maximum amount of ash content (%) was also observed in the red + black seed colored accession, i.e., IC0418119 (4.01%). From Table 3, it can be noted that IC0418119 (4.01%), IC0310855 (4.00%) and IC0469946 (3.95%) were the top three accessions with the highest amount of ash content. The mean value of ash content (%) among the different seed colored *Abrus precatorius* accessions studied is shown in Figure 5.

### 2.6. Grouping Comparison with ANOVA and Further Post Hoc Tests (Tukey’s HSD Test)

All the 99 accessions were categorized into five groups based on seed color, i.e., complete black, red + black, complete white, cream + brown and pink + brown. The statistical variation among and within groups was studied using ANOVA and further post hoc tests (Tukey’s HSD test), which is shown in Table 4. Based on the analysis, it was found that highly significant variation exists at a 1% level of significance between various groups for biochemical parameters, like total monomeric anthocyanin, flavonols and antioxidant content. For protein content and ash content, there exists no significant variation between the groups. Similarly, the variation within the group for all biochemical parameters was non-significant.

The mean total monomeric anthocyanin of the different color groups is shown in Table 5. It was observed that completely black colored accessions differ significantly from the other groups. Similarly, red + black accessions differ significantly from the other groups. All the remaining color groups, viz. complete white, cream + brown and pink + brown, do not show significant differences. The mean flavonol content of different color groups is shown in Table 6. It was noted that completely black colored accessions differ significantly from other groups. Similarly, red + black accessions differ significantly from the other groups, but the pink + brown and cream + brown accessions were statistically on par. Similarly, cream + brown and complete white accessions were statistically on par. The mean antioxidant content of the different color groups is shown in Table 7. It was found that complete black and red + black accessions were statistically on par. Similarly, pink + brown, cream + brown and complete white were statistically on par.

### 2.7. Correlation Analysis

The correlation analysis was carried out to study the association among various biochemical parameters in *Abrus* germplasm. The result of the correlation analysis is presented in Figure 6 (Software used: R studio version 4.2.3.) [16]. Based on the scale, the degree association between various traits was studied [17]. There exists a highly significant, very strong and positive association between antioxidants and anthocyanin (r = 0.92), antioxidants and flavonols (r = 0.87) and anthocyanin and flavonols (r = 0.92). On the contrary, the association among the other parameters studied was found to be non-significant. Previous studies have reported that the antioxidant activity of *Abrus* extracts was positively correlated with the total phenol content. It was also reported that the total phenol content was associated with total flavonoid content [18]. Similarly, a strong correlation between total phenol content, total flavonoid content and their antioxidant activities in *Abrus cantoniensis* and *Abrus mollis* was reported [19].

## 3. Discussion

Previous studies on the seed composition of *Abrus precatorius* were very limited. The average values of the total monomeric anthocyanin content among complete black and red + black accessions were 49.8 and 27.68 mg/100 g of CGE. Previous studies have reported 60.44 ± 0.50 CGE/100 g as the total anthocyanin content from the crude extract of *A. precatorius* seed coat [20]. The mean flavonol content of the complete black, complete white, cream + brown, pink + brown and red + black seeded accessions were recorded as 63.54 mg/g, 26.84 mg/g, 27.66 mg/g, 33.28 mg/g and 44.98 mg/g of QE, respectively. Most of the previous studies were focused on the estimation of total flavonoid content of the leaves. It was reported that the total flavonoid content of different extracts of *Abrus* leaves, viz. ethanol, water, and petroleum ether, were 14.43 ± 1.35, 20.84 ± 1.97, 1.6 ± 0.41 mg/g of QE dry extract [21]. The hydro-methanolic extract of the seed contains the total flavonoid content of 73.33 ± 2.36 mg/g of a rutin equivalent (RE) [22]. The mean antioxidant content of the complete black, complete white, cream + brown, pink + brown and red + black seeded accessions were recorded as 13.6 mg/g, 1.13 mg/g, 1.96 mg/g, 2.37 mg/g and 12.27 mg/g of GAE, respectively. Previous researchers have observed that the antioxidant capacity of methanolic leaf extracts was at maximum, with a low IC50 value of 62.86 ± 0.68 µg/mL [23]. The leaf extract was found to have an antioxidative potential ranging between 2.67 ± 0.40 to 13.34 ± 0.35 mg/g of ascorbic acid equivalents (AAE) [14]. The previous studies have mentioned its pharmacological potential by estimating the polyphenol content [18,20,21,22,24]. The mean protein content of the complete black, complete white, cream + brown, pink + brown and red + black seeded accessions were recorded as 18.10%, 18.11%, 17.84%, 18.54% and 17.96%, respectively. The protein content of the *Abrus* seed was previously studied by a few researchers using the Kjeldahl method [13]. Their study revealed the protein content as 16.28% which was closer to that of Australian pulse crop, *Cassia notabilis*. Some researchers have studied the crude protein content in the leaves of *Abrus precatorius*. Based on their study, the protein content was estimated as 8% in the leaves [23]. The mean ash content of the complete black, complete white, cream + brown, pink + brown and red + black seeded accessions were recorded as 3.27%, 3.40%, 3.13%, 3.12% and 3.27%, respectively. It was reported that the ash content of *Abrus* seeds was 3.36% [13]. They also mentioned that the value of ash content was similar to that of pinto bean (*Phaseolus vulgaris*) and Narrowleaf lupin (*Lupins angustifolius* and *L. hispanicus*). Some early investigations were carried out on the ash content of *Abrus* leaves. Based on their study, the amount of ash content in *Abrus* leaves was 7.00 ± 1.41% [23]. *Abrus pulchellus* and *A. precatorius* differ from each other in terms of their ash content. The former was recorded as 3.75%, whereas the latter was recorded with only 2.75% [25]. A few earlier studies have estimated the ash content in its different forms, viz. acid insoluble ash, water soluble ash and sulphated ash content [26,27,28]. The amount of acid insoluble ash, water soluble ash and sulphated ash in *Abrus* root powder was estimated as 0.1%, 0.2% and 0.08%, respectively [26]. The total ash content in the *Abrus* roots was estimated as 7.01% [27]. The acid insoluble ash content and water soluble ash content of *Abrus* seeds were found to be 1.0% and 3.17%, respectively [28]. A grouping comparison with ANOVA and further post hoc tests (Tukey’s HSD test) revealed that for all the three biochemical parameters, complete black and red + black differ significantly from the other color groups. Accessions with different seed colors vary significantly in terms of their bioactive constituents. The present investigation provides experimental proof that *Abrus* seeds contain considerable amounts of total monomeric anthocyanin (except complete white, cream + brown and pink + brown colored accessions), flavonols and antioxidants.

The findings of the experiment were most reliable, as it was conducted using the accessions representing the entire Indian sub-continent (except north eastern India). Though the focus of other studies was only on red + black seed color, this experimental result represents the entire genetic variability present in the region. Most of the previous research work was carried out on its leaves, but the present work was carried out on the seeds with different color combinations. However, the major limitation of this research was the fact that the exotic collections (accessions from regions other than India) were not included in the study (as they were not present in the Indian National Genebank). Clinical trials to understand their mechanisms were not conducted in the present study.

## 4. Materials and Methods

### 4.1. Seed Material

A total of 99 accessions were chosen based on germination % and seed quantity from the Indian National Genebank, National Bureau of Plant Genetic Resources (NBPGR). These accessions represent the different agro-ecological zones of India, with the exception of north east India. Seeds from the Genebank were planted at two locations, which were NBPGR regional station, Ranchi, and NBPGR experimental farm, Issapur, and the derived seeds were used for various tests. The five main groups, based on the seed color (as shown in Figure 7) and their corresponding RHS color code, were as follows: red + black combination (43A + 202A), pink + brown combination (65D + 199D), cream + brown combination (156D + 199B), complete black (202A + 42B) and complete white color (155C). The number of accessions present in the red + black combination, the cream + brown combination, the pink + brown combination, as well as complete black and complete white colors, were seventy-seven, five, two, three and twelve respectively. All the samples were indigenous collections (IC). The list of accessions, seed color, and their passport data were given in Appendix A (Table A1).

### 4.2. Estimation of Total Monomeric Anthocyanin Pigment Content by pH Differential Method

This method was based on the color change in monomeric anthocyanin, reversible with a change in pH. The materials required are standard buffer solutions, pH meter—standardized with pH 4.0 and 7.0 (RC-12, Radicon, Surat, India), volumetric flasks (50 mL), a spectrophotometer (XD 7500, BenchTop Lab Systems, Saint Louis, MO, USA), and a cuvette (1 cm path length).

#### 4.2.1. Reagents Required

Potassium chloride (0.025 M) (Anish chemical, Bhavnagar, India, 99.5%, CAS: 07-09-7440): Weigh 1.86 g KCl and transfer it into a beaker. Then, add 980 mL distilled water. Adjust the pH to 1.0 with HCl (Merckmillipore, Bangalore, India, 37%, 7647-01-0), then transfer it to a volumetric flask of 1 L and make up the required volume using distilled water. Sodium acetate (0.4 M) (Vinpul chemicals, Vadodara, India, 99%, 127-09-3): For this, measure 54.43 g of sodium acetate trihydrate and add 960 mL of distilled water. Adjust the pH to 4.5 with HCl, and then transfer it to a volumetric flask of 1 L. Finally, make up the required volume using distilled water.

#### 4.2.2. Procedure for Estimation of Total Monomeric Anthocyanin Pigment Content

Weigh 0.1 g of the sample and transfer it into a 15 mL centrifuge tube. Add 10 mL of 0.1 N HCl into the tube and keep it in rotospin (Cat: 3090 Rotospin rotary mixer, Dhana foundations and technology (DFT), Chennai, India) overnight so that the contents mix properly. After that, centrifuge the tubes at 10,000 rpm for 10 min. Pipette out 500 µL of supernatant in two different glass tubes. Then, transfer 1500 µL of HCl in one tube and 1500 µL of sodium acetate in another tube and mix the contents to prepare the buffer. Blank 1 for pH 1.0 buffer: pipette out 500 µL of 0.1 N HCl into a glass tube and add 1500 µL of HCl into it. Blank 2 for pH 4.5 buffer: pipette out 500 µL of 0.1 N HCl into a glass tube and add 1500 µL of sodium acetate into it. Keep them in room temperature for 30 min and vortex them to mix the contents. Take an absorbance reading at 520 and 700 nm. If the test portion is turbid, then centrifuge it to clarify them. Another method is to use a filter (Millipore TM membrane filter, 1.2 mm pore size). The unit for measuring the total monomeric anthocyanin was mg/100 g of cyanidin 3-glucoside equivalent (CGE). Anthocyanin content was calculated using the formula below:Anthocyanin = A × MW × DF × v × 102/ε × l × W
where

A (Absorbance) = (A520 nm–A700 nm) at pH 1.0–(A520 nm–A700 nm) at pH 4.5;MW (Molecular weight) = 449.2 g/mol;DF = Dilution factor;l = Path length in cm;ε = 26,900 molar extinction coefficient (M^−1^cm^−1^);W—Weight of sample (g) andV—Volume of test solution (mL)

### 4.3. Flavonol Estimation by OOMAH Method

This involves the formation of an acid stable and acid labile complex formed by flavonol, which has maximum absorbance at 360 nm [29].

#### 4.3.1. Reagents Required

The following reagents were required for flavonol estimation: 75%, 80% and 95% ethanol (Merckmillipore, Bangalore, India, 99.9%, 64-17-5), 2% HCl (Merckmillipore, India, 37%, 7647-01-0) in 75% ethanol and the quercetin standard (Lobachemie, Mumbai, India, 95%, 849061-97-8). The stock concentration of quercetin should be 1 mg/mL. This can be prepared by dissolving 25 mg of quercetin in 25 mL of 95% ethanol. The working standard can be prepared by pipetting 4 mL from the stock solution and then making up the volume to 50 mL by using 95% ethanol.

#### 4.3.2. Procedure for Flavonol Estimation

The sample (200 mg) was extracted with 80% aqueous ethanol (10 mL). The samples were centrifuged (10,000 RPM for 10 min) and the supernatants were recovered. A total of 100 μL of the sample was placed in a test tube. Add 2.4 mL of 2% HCl in 75% ethanol. Prepare the standard with different concentrations (16, 32, 48, 64, 80 μg/mL) and blank. These were labeled as S1, S2, S3, S4, S5, and then blanked. The working solution of quercetin and 80% ethanol was used to prepare standards of different concentrations. In the blank, working solution was not added, i.e., the 100 μL were made up with 80% ethanol. The sample and standards were prepared at the same time. Stir the contents by using a vortex and keep it for 30 min at room temperature. Then, measure the absorbance at 360 nm by using a spectrophotometer (XD 7500, BenchTop Lab Systems, St. Louis, MO, USA). The spectrophotometer readings for different standards were recorded and the values were plotted in a graph. Then, the spectrophotometer readings for the samples were noted down. Based on the graph, the amount of flavonol in the samples was estimated. The results were expressed as mg/g of the quercetin equivalent (mg/g of QE) of the sample.

### 4.4. Estimation of Antioxidants by Ferric Reducing Antioxidant Potential (FRAP)

The FRAP method is based on the reduction potential of complex Ferric-tripyridyltriazine (Fe^3+^-TPTZ) to a blue-colored Ferrous-tripyridyltriazine (Fe^2+^-TPZ) complex form by antioxidants at a lower pH [30].

#### 4.4.1. Reagents

The reagents required were as follows: 0.1 M acetate at pH 3.6. This can be prepared by dissolving 8.2 g of sodium acetate (Vinpul chemicals, India, 99%, 127-09-3) in 1 L of distilled water. The 20 mM Ferric chloride (FeCl_3_·6H_2_O) can be prepared by dissolving 8.1 mg Ferric chloride (Alphachemika, Mumbai, India, 98%, 7705-08-0) in 2.5 mL of distilled water. The 10 mM Tripyridyltriazine (TPTZ) can be prepared by dissolving 8.34 mg of TPTZ (SRL chemicals, Mumbai, India, 99%, 3682-35-7) in 2.5 mL of 40 mM HCl (Merckmillipore, India, 37%, 7647-01-0). The Gallic acid (Oxford Lab Fine Chem LLP, Navghar, India, 99.5, 5995-86-8) was used as the standard stock solution, which was prepared by dissolving 0.1 g Gallic acid in 100 mL of distilled water. The working standard 1 was prepared by pipetting 10 mL of stock solution and then making up the volume to 100 mL. After that, working standard 2 was prepared by pipetting 10 mL of working standard 1 and then making up the volume to 100 mL. From working standard 2 take 50 µL, 100 µL, 150 µL, 200 µL, 250 µL and 300 µL and make up their volume to 300 µL by using distilled water. The blank was prepared with 300 µL of distilled water. FRAP reagent was freshly prepared by combining the above reagents (acetate, TPTZ solution and ferric chloride) in 10:1:1 ratio.

#### 4.4.2. Procedure for Antioxidant Estimation

A total of 0.1 g of sample was added into 5 mL of 80% ethanol. The contents were mixed properly by using vortex. Centrifuge the content at 10,000 RPM for 10 min and collect the supernatant. The volume of sample required varies among the accessions with different seed color, i.e., 20 µL of red + black colored accessions, 20 µL completely black colored accessions and 100 µL of white colored accessions. Add 2200 µL of freshly prepared FRAP reagent to the sample taken in test tubes. Make up the final volume to 2500 µL (i.e., 280 µL of distilled water for red + black colored accessions, 280 µL for completely black colored accessions and 200 µL for white colored accessions). Then, vortex each test tube and incubate it at room temperature for 30 min. During incubation it turns an indigo blue color. Note down the absorbance reading by using a UV spectrophotometer (XD 7500, BenchTop Lab Systems, St. Louis, MO, USA) at 593 nm. Prepare standards (S1, S2, S3, S4 and S5) and blank. The unit for measuring antioxidant activity was mg/g of gallic acid equivalent (GAE)

### 4.5. Determination of Protein Content by Dumas Method

This method was based on the determination of total nitrogen content in the sample of organic matrix via subsequent oxidation and reduction by Dumas instrument (Analysensysteme GmbH, Elementar, Langenselbold, Germany). The sample is combusted at very high temperature (700–1000 °C) in the presence of oxygen. The gasses are produced, in which the CO_2_ is trapped and the remaining gasses are reduced by copper. The nitrogen produced will be detected by a thermal conductivity detector. The protein conversion factor used in this method was 6.25.

#### Procedure for Determination of Protein Content

Inject the samples into a combustion tube (940 °C) along with oxygen. The gathering ring in the combustion tube increases the temperature up to 1800 °C during combustion. This results in oxidation and halogen trapping (silver cobalt and chromium sesquioxide). This was followed by reduction (nitrogen oxides into nitrogen by copper at 700 °C). The chromatography helps in the separation of nitrogen and methane. Unmeasured elements like H_2_O and CO_2_ are trapped using anhydrone and ascarite, respectively. After chromatography, the detection of nitrogen was carried out by catharometer and further data processing took place. The procedures involved were as follows: Weigh 0.1 g of sample and place it properly at the hole position in the instrument. Aspartic acid (Alchemax, Hyderabad, India, 98%, 1783-96-6) was used as the standard. Properly close the lid and run the samples. Then, the instrument will burn the sample and record the protein content per 100 g of sample.

### 4.6. Estimation of Ash Content by Gravimetric Method

Ash is an inorganic residue that remains after the removal of water and organic matter in the presence of oxidizing agents by heat. It helps in measuring the total minerals within the sample, and the unit was gram of ash per 100 g sample.

#### 4.6.1. Materials Required

4 N Nitric acid (Choice Organochem, Hyderabad, India, 98%, 7697-37-2), Furnace (BST/MF/900, Bionics Scientific, New Delhi, India), an analytical balance (Model ATX224R, Shimadzu, Kyoto, Japan), an air oven (LBS-AG-1, Labsol Enterprises, Gurugram, India), tongs, a spatula and a silica dish.

#### 4.6.2. Procedure for Estimation of Ash Content

Initially, keep the crucible in the furnace for 2–3 h at 500–550 °C. After some time, transfer the crucible to the oven (100 °C for 1–2 h) to reduce the temperature of crucible to 100 °C, and the weight should be noted immediately as *W*_1_. Again, take 2–4 g dry samples in replicates in the crucible and note it as *W*_2_. For the burning of the sample, place the crucible that contains the samples in a furnace at 180 °C for 1 h and gradually increase the temperature up to 250 °C for 1 h. Maintain the samples in the furnace for 3 h at 450 °C. This temperature is achieved by gradually increasing temperature (50 °C) at a 30 min interval from 180 °C to 450 °C. Check whether the sample is completely dried to a white-colored ash form or not. If not properly dried, then add concentrated nitric acid (2–3 drops). After that, evaporate it in water bath and repeat the same heating procedure in the furnace (30–60 min) until the stable reading is achieved. Again, decrease the temperature of the furnace and take the weight (*W*_3_) at 100 °C. The ash content was measured by g per 100 g.
$$\mathrm{Ash}\;\mathrm{content}=\frac{\left(W_{3}-W_{1}\right)}{\left(W_{2}-W_{1}\right)}\times 100$$

*W*_1_—weight of crucible (g),*W*_2_—weight of crucible + sample (g) and*W*_3_—weight of crucible + ash (g).

### 4.7. Statistical Analysis

The data were evaluated by a one-way analysis of variance, and comparison was carried out by least significant difference (LSD) by using the software WASP 2.0 (Web Based Agricultural Statistics Software Package). A grouping comparison with ANOVA and further post hoc tests (Tukey’s HSD test) was carried out using SPSS Statistics v29. If the *p*-value was <0.05, there exists a statistically significant difference among the accessions. All the 99 treatments or accessions were replicated three times and the final results were shown as the mean values. Correlation analysis was performed by using the Karl Pearson correlation coefficient and the software used was R studio version 4.2.3.

## 5. Conclusions

Herbal medicines are gaining momentum among the people, as they cause no or less side effects than modern drugs. Based on the present investigation, the potential of *Abrus precatorius* in curing degenerative diseases was higher in completely black colored and red + black colored accessions. The superior accessions identified from the experiment hold huge pharmaceutical potential. The pharmaceutical companies can develop drugs which can be used for the treatment of diseases resulting from oxidative stress. In the traditional systems of medicine, like Siddha and Ayurveda, the practitioner can make use of these elite accessions identified. Future research can focus on the specific mechanisms responsible for such variations and also develop effective methods for removing the toxic constituents without affecting their pharmacological value. Proper clinical trials should be conducted for their therapeutic implications.

## Figures and Tables

**Figure 1 plants-13-01963-f001:**
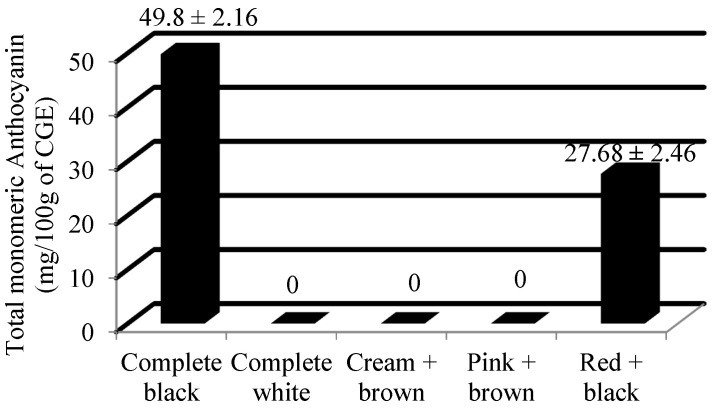
Mean value ± Standard deviation of total monomeric anthocyanin (mg/100 g of CGE) among different seed colored *Abrus precatorius* accessions.

**Figure 2 plants-13-01963-f002:**
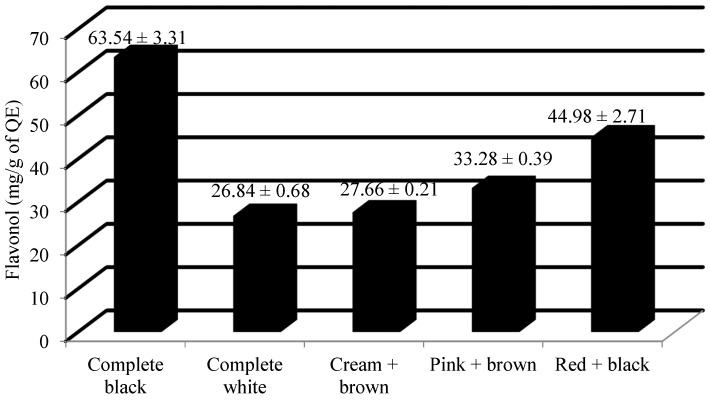
Mean value ± Standard deviation of flavonol (mg/g of QE) among different seed colored *Abrus precatorius* accessions.

**Figure 3 plants-13-01963-f003:**
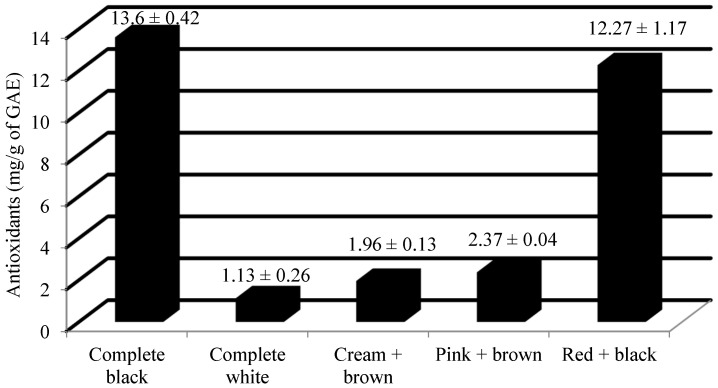
Mean value ± Standard deviation of antioxidants (mg/g of GAE) among different seed colored *Abrus precatorius* accessions.

**Figure 4 plants-13-01963-f004:**
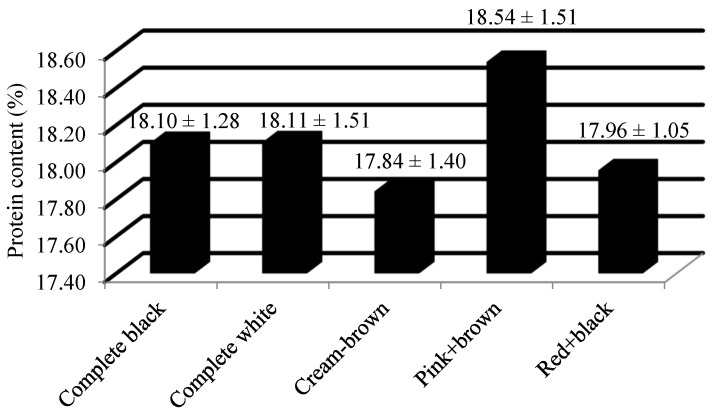
Mean value ± Standard deviation of protein content (%) among different seed colored *Abrus precatorius* accessions.

**Figure 5 plants-13-01963-f005:**
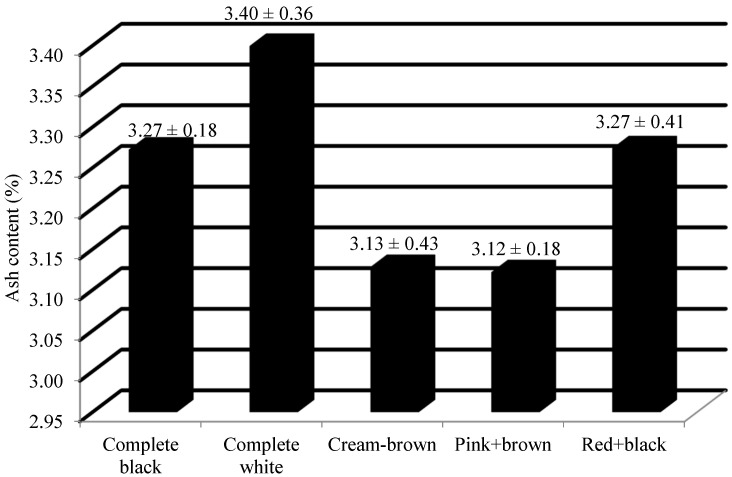
Mean value ± Standard deviation of ash content (%) among different seed colored *Abrus precatorius* accessions.

**Figure 6 plants-13-01963-f006:**
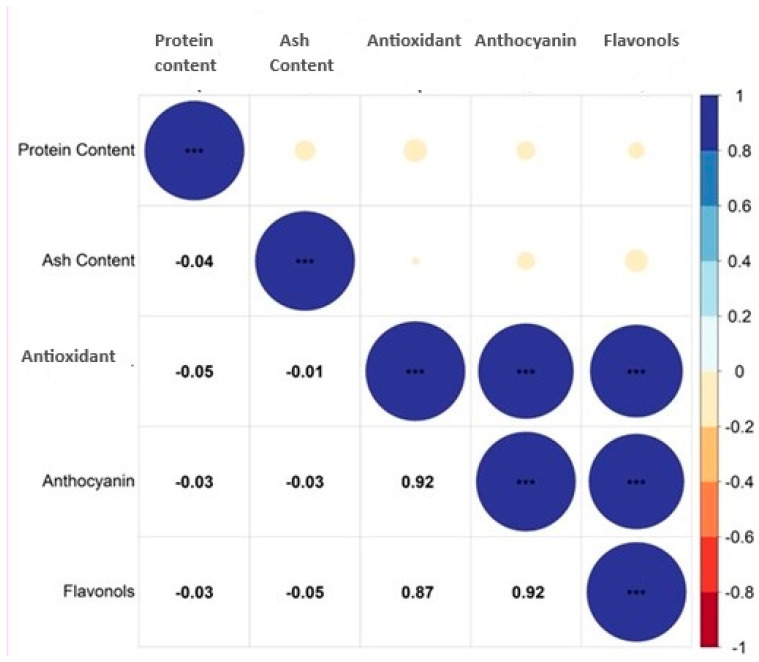
Correlation matrix depicting the relationship between various chemical components of seeds in *Abrus precatorius*. *** (in the diagnol) denotes same parameters; *** (other than the diagnol) indicates highly significant correlation among various biochemical parameters.

**Figure 7 plants-13-01963-f007:**
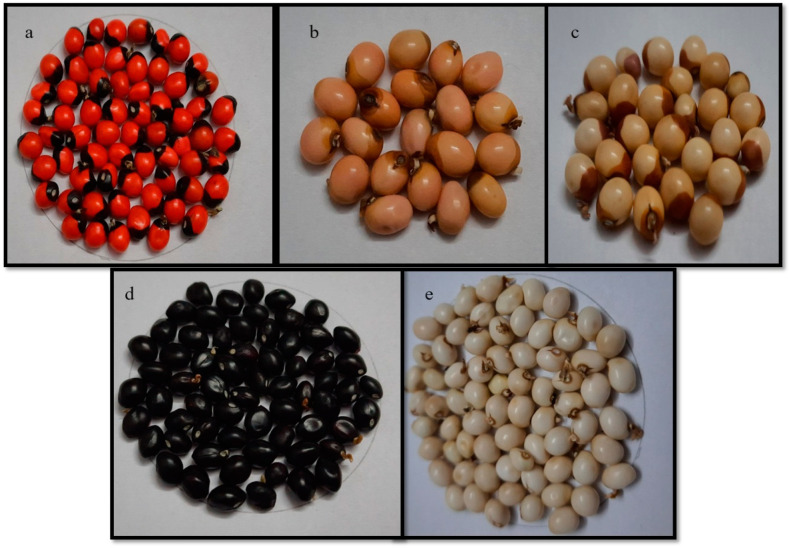
Seed coat color diversity among the *Abrus precatorius* germplasm conserved at the Indian National Genebank. (**a**) Red + black combination; (**b**) pink + brown combination; (**c**) cream + brown combination; (**d**) complete black color and (**e**) complete white color.

**Table 1 plants-13-01963-t001:** Completely randomized design (CRD) ANOVA based on biochemical parameters.

Source	df	Anthocyanin	Flavonol	Antioxidant	Protein	Ash
Treatment	98	293.15 **	134.58 **	39.16 **	2.51 **	0.31 **
Error	99	9.39	23.52	0.08	2.14	0.43
Total	197	-	-	-	-	-
Critical difference (CD)	-	8.06	12.75	0.76	3.84	1.71

** indicates highly significant variation at a 1% level of significance.

**Table 2 plants-13-01963-t002:** Descriptive statistics of *Abrus precatorius* accessions on the basis of biochemical parameters.

Descriptive Statistics	Total Monomeric Anthocyanin (mg/100 g of CGE)	Flavonols (mg/g of QE)	Antioxidants (mg/g of GAE)	Protein Content (%)	Ash Content (%)
Mean	23.04	42.23	10.24	17.99	3.28
Standard Error	1.22	0.83	0.45	0.11	0.04
Median	27.07	44.04	11.65	18.03	3.34
Range	51.95	40.80	13.42	5.23	1.93
Minimum	0.00	25.61	0.76	15.43	2.08
Maximum	51.95	66.41	14.18	20.67	4.01

**Table 3 plants-13-01963-t003:** The top three best performing accessions in *Abrus precatorius* accessions based on phytochemical analysis.

S. No.	Biochemical Parameters	Best Performing Accessions
1	Total monomeric anthocyanin (mg/100 g of CGE)	IC0605143 (51.95 mg/100 g), IC0405311 (49.83 mg/100 g) and IC0401666 (47.63 mg/100 g)
2	Flavonols (mg/g of QE)	IC0405311 (66.41 mg/g), IC0401666 (64.31 mg/g) and IC0605143 (59.91 mg/g)
3	Antioxidants (mg/g of GAE)	IC0405295 (14.18 mg/g), IC0385638 (14.13 mg/g) and IC0405311 (14.07 mg/g)
4	Protein content (%)	IC0385644 (20.67%), IC0538733 (20.41%) and IC0349819 (20.17%)
5	Ash content (%)	IC0418119 (4.01%), IC0310855 (4.00%) and IC0469946 (3.95%)

**Table 4 plants-13-01963-t004:** Grouping comparison with ANOVA.

Source of Variation	SS	df	Anthocyanin	Flavonol	Antioxidant	Protein	Ash
Between group	231.78	4	258.76 **	127.56 **	31.63 **	2.16 ^ns^	0.46 ^ns^
Within group	289.64	94	8.17 ^ns^	21.87 ^ns^	0.12 ^ns^	2.08 ^ns^	0.51 ^ns^
Total	521.42	98					

The denotation ** indicates the highly significant difference at a 1% level of significance and ^ns^ denotes no significant difference among the groups.

**Table 5 plants-13-01963-t005:** Grouping information of based on anthocyanin by using the Tukey HSD test.

Different Color Group	Mean Anthocyanin in Decreasing Order (mg/100 g of CGE)	Grouping Information Based on Significance
Complete black	49.80	A
Red + black	27.68	B
Complete white	0.00	C
Cream + brown	0.00	C
Pink + brown	0.00	C

**Table 6 plants-13-01963-t006:** Grouping information of based on flavonols by using the Tukey HSD test.

Different Color Group	Flavonol Mean in Decreasing Order (mg/g of QE)	Grouping Information Based on Significance
Complete black	63.54	A
Red + black	44.98	B
Pink + brown	33.28	C
Cream + brown	27.66	C D
Complete white	26.84	D

**Table 7 plants-13-01963-t007:** Grouping information of based on antioxidants by using the Tukey HSD test.

Different Color Group	Antioxidants Mean in Decreasing Order (mg/g of GAE)	Grouping Information Based on Significance
Complete black	13.60	A
Red + black	12.27	A
Pink + brown	2.37	B
Cream + brown	1.96	B
Complete white	1.13	B

## Data Availability

All data are contained within the article. Additional details are available from the corresponding author, P.S., upon reasonable request.

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
