# Peer review of "Evaluation of Chemical Composition among the Multi Colored Germplasm of Abrus precatorius L."

_plants, 2024, doi:10.3390/plants13141963_

Round 1

Reviewer 1 Report

Comments and Suggestions for Authors

I kindly ask you to make the following additions:

1. What nitrogen to protein conversion factor was used in the Dumas method.

2. There is no such thing as HCl buffer and sodium acetate buffer, I think the authors used a mental shortcut. A buffer is a mixture of a weak acid and a salt of that acid with a strong base, or a mixture of a weak base and a salt of that base with a strong acid. Please correct this.

3. There are misspellings of units in the text, it should be written mg QE/g, not (mg/g of QE).

Once the additions have been made (in the points above), the paper may be published in a journal Plants.

Reviewer 2 Report

Comments and Suggestions for Authors

Thank you for yours instructive articel. I have some coments and proposal.

I miss the expalatation of the acronyms. For example 43D+202A, DFT, IC0349819,... .

I would be reorder the chaptures: Introduction, Materials and methods, Results, Dicusion, conclusions.

Some Tittel in chapture "Material and Methods" should be to better describe. Procedure? How procedure? For example: procedure of estimation of flavonols. Subchaptures in "material and Methods" should be to have the same breakdown.

Formulas should be to wrtiting in Equation Editor. The quantity Molar extinction has wrong unit notation. Other quantities in formulas don´t have any units? They have, please to list them.

I don´t understant to using statistic, You measured three samples and than from they you determined statistic.

References don´t have DOI and all is with acronym "[CrossRef]".
